# Identification and Characterization of the *AREB*/*ABF*/*ABI5* Gene Family in Sandalwood (*Santalum album* L.) and Its Potential Role in Drought Stress and ABA Treatment

Xiaojing Liu [1,2,†], Renwu Cheng [3,†], Yu Chen [1], Shengkun Wang [2], Fangcuo Qin [2], Dongli Wang [2], Yunshan Liu [2], Lipan Hu [1,*] and Sen Meng [1,2,*]

1   College of Biology and Food Engineering, Chongqing Three Gorges University, Wanzhou, Chongqing 404100, China; lxj15025545830@163.com (X.L.); cheny8051005@163.com (Y.C.)
2   State Key Laboratory of Tree Genetics and Breeding, Research Institute of Tropical Forestry, Chinese Academy of Forestry, Guangzhou 510520, China; wskun2001@163.com (S.W.); qinfc@caf.ac.cn (F.Q.); wangdongli1997@163.com (D.W.); lys061949@163.com (Y.L.)
3   Guangzhou Institute of Forestry and Landscape Architecture, Guangzhou 510520, China; crw_cheng@163.com
*   Correspondence: xinonghulipan@163.com (L.H.); mengsen021124@126.com (S.M.)
†   These authors contributed equally to this work.

**Abstract:** *AREB/ABF/ABI5* (ABA-responsive element-binding protein/ABRE binding factors and ABA INSENSITIVE 5) transcription factors are involved in regulating the expression of ABA (abscisic acid)-related genes and improving plant adaptability to environmental stress. To explore the influence of *AREB/ABF* transcription factors on santalol synthesis, we conducted a genome-wide analysis of the *AREB* gene family in sandalwood, identified 10 *SaAREB* genes, and divided them into five subfamilies. We found that all *SaAREB* genes encoded unstable hydrophilic proteins and the subcellular localization prediction of *SaAREB*s was that they are located in the nucleus. *AREB/ABF* genes belong to the bZIP-A subfamily and we found that the 10 AREB proteins all contained bZIP (basic region leucine zipper) and four potential phosphorylation sites (RXXS/T). According to the collinearity analysis results, four of the *SaAREB* genes were involved in two fragment duplication events. Through qRT-PCR (real-time fluorescence quantitative PCR), we explored the expression profile of *SaAREB* in different tissues; the effects of ABA treatment and drought treatment on *AREB* transcription factors were predicted. From the expression of different tissues, we found that *SaAREB1* not only responded to prolonged drought but also was highly expressed in stems. Moreover, *SaAREB3*, *SaAREB7*, and *SaAREB8* specifically respond to ABA treatment. Based on RNA-seq (RNA sequencing) data, we found that *SaAREB6* and *SaAREB8* were highly expressed in the sapwood and transition regions. Regarding *SaCYP736A167*, as a key gene in santalol synthesis, its promoter contains the most ABRE cis-reactive elements. These results provide a basis for further analysis of the role of the *Santalum album* L. (*S. album*) *ABRE/ABF/ABI5* genes in the formation of santalols.

**Keywords:** *AREB/ABF/ABI5*; *SaCYP736A167*; terpenoids; drought; RNA-seq

## 1. Introduction

Sandalwood is a semiparasitic plant of the genus Santalum in the family Santalaceae, which has high commercial and medicinal value [1]. Sandalwood heartwood, as an important raw material used to produce essential oils and spices, has always had a high market heat. In terms of sandalwood, the heartwood contains a unique aroma; it is mainly used in the manufacture of high-end furniture and the essential oil extracted from sandalwood heartwood has increasingly become an important raw material for the cosmetics and pharmaceutical industries [2,3]. At the same time, sandalwood, as a medicinal herb, has significant effects on anticancer, antioxidant, and neurological aspects. The sesquiterpene

alcohols α-santalol, β-santalol, epi-β-santalol, and α-exo-bergamotol are the indispensable components of *S. album* essential oil [4]. In recent years, terpenoids have been playing more and more important roles in chemistry, drugs, cosmetics, etc., and the demand for terpenoids has been increasing. Meanwhile, we find a large amount of the literature has reported the effects of environmental factors on the synthesis of terpenoids in plants [5].

As one of the most important environmental factors, drought-induced osmotic stress significantly influences the synthesis of antioxidants, flavonoids, and secondary metabolites in the plants [6]. When plants are stressed, the balance of metabolism will be destroyed; plants may adapt to environmental changes by adjusting metabolic pathways and the release of terpenes can help plants resist stress.

Based on relevant studies, the synthesized terpenoids of plants, mainly through MEP and MVA pathways, and drought stress mainly change the carbon supply in the MEP pathway by affecting the photosynthetic rate and stomatal conductance before affecting the production of terpenes [7]. Additionally, 3-Hydroxy-3-methylglutarylcoenzyme A reductase (HMGR) is the rate-limiting enzyme in the MVA pathway; drought can significantly affect the expression of HMGR and, thus, affect the accumulation of tanshinone [8]. Similarly, we found that the sesquiterpene alkenes and alcohols in sandalwood essential oil were mainly generated through the MVA (mevalonate) pathway. Santalene synthase (*SaSSY*), a key gene in the downstream part of the MVA pathway, can catalyze (E, E)-FPP to generate α-santalene, β-santalene, α-exo-bergamotene, and epi-β-santalene [9]. It was further catalyzed by *SaCYP736A167*, which belonged to the cytochrome P450 gene family, to produce santalol [10]. Through preliminary analysis of the promoter of this gene and the statistics of cis-reaction elements, we found that there were a large number of ABREs (ABA-responsive elements). In exploring the influence of *AREB/ABF/ABI5* transcription factors on the formation of sandalwood alcohol, we identified the *AREB/ABF/ABI5* gene family of sandalwood.

*AREB/ABF/ABI5* transcription factors are basic leucine zipper proteins and their functions are involved in the plant response to hormones and the ability to influence stress [11,12]. Meanwhile, ABA can be used as a response signal to drought stress. Under drought stress, plants accumulate more ABA, which, in turn, preserves water by affecting stomatal closure [13,14]. Meanwhile, drought-induced osmotic stress significantly influences the synthesis of secondary metabolites in plants [15–18]. For example, moderate drought can promote the release of plant terpene in Holm Oak (*Quercus Ilex* L.), spearmint (*Mentha spicata* L.), and rosemary (*Rosmarinus officinalis* L.). However, this effect may also depend on stress severity and stress duration.

*AREB/ABF/ABI5* is a *bZIP*-like transcription factor that binds to ABRE and activates the expression of ABA-dependent genes under drought stress [19]. Similarly, the role of the cis-reactive element ABRE in ABA-response gene expression has been characterized in detail [20]. The ABA-responsive element (ABRE, ACGTGGC) in sandalwood is a representative of the conserved sequence PYCGTGGC [21,22]. ABRE is similar to the G-box (CACGTG), which is present in the promoters of light-regulated genes and it is necessary for the expression of ABA-induced genes [23]. ABA induction usually requires additional elements. These elements, often called "coupling elements", together with G-ABRE, form ABA-reaction complexes [24,25]. The isolated interactors that have an ABRE sequence basically belong to the bZIP class or can be combined with them in vitro [26,27]. In the same way, a class of *DREB* genes, which are AP2/ERF proteins, can also be used as coupling elements with ABREs to induce the expression of ABA-dependent genes [28]. In terms of conserved domains, the N-terminus of AREB contains C1, C2, and C3; C4 is located at the C-terminus of the protein. Meanwhile, phosphorylation sites (RXXS/T) are present in all four conserved domains. In addition, the C-terminus also contains a highly conserved alkaline leucine zip structure that binds DNA and other proteins when activated by phosphorylation [29].

In previous studies, we find there are nine *AREB/ABF/ABI5* subfamily genes in *A. thaliana* (*Arabidopsis thaliana*): *AtABF1*, *AtABF2/AREB1*, *AtABF3*, *AtABF4/AREB2*, *AtABI5/DPBF1*, *AtDPBF2*, *AtDPBF3/AREB3*, *AtDPBF4*, and *AtbZIP15* [30,31]. *ABF1* has been shown to be a type of gene that responds to cold and low-temperature stress while the other three *AREB/ABFs* mainly respond to ABA- and osmotic-pressure-related stress [32]. Among them, *ABIs* are associated with ABA signal transduction during seed maturation [33]. However, research has found that after abiotic stress, *AREB/ABFs* have higher expression in vegetative tissues compared to other tissues. *ABI5* plays a key role in the regulation of ABA signaling and stress responses in *A. thaliana* seedlings because ABA induces the expression of *ABI5* and modifies its phosphorylation status [34]. *SlAREB1*, the homologous gene of *AREB1*, can improve the activity of antioxidant enzymes and their defense power against environmental stress by directly interacting with *SlMn-SOD* [35].

As a transcription factor that specifically binds to ABRE cis-reactive elements, *AREB* can respond to drought stress and moderate drought can affect the formation of terpenoids. In addition, *SaCYP736A167*, as a key gene in santalol synthesis, contains a large number of ABRE cis-reaction elements in its promoter; so, we can reasonably predict that *AREB* may be related to the formation of sandalwood terpenoids. To preliminarily understand the functions of *AREB/ABF* in *S. album* and whether *AREB* can affect the formation of sandalwood terpene compounds, all of the members of the *AREB* family in sandalwood were identified using the latest sandalwood genome and a total of 10 *SaAREB* genes were screened. We then performed detailed investigations on the physicochemical properties of *SaAREB*, involving phylogenetic analysis, gene structure, conserved motifs, chromosome localization, and collinear relationships. We also analyzed the expression patterns of *SaAREB* in roots, leaves, haustoria, phloem, sapwood, and transition zones using RNA-seq data. To study the expression pattern of the *SaAREB* gene in different tissues under ABA and drought stress, we used real-time fluorescence quantitative PCR (qRT-PCR) and analyzed the relative expression. This study will help to identify the function of sandalwood *AREB* transcription factors and provide important candidate genes for the formation of sandalwood heartwood.

## 2. Materials and Methods

### 2.1. Plant Material and Treatment

Five-month-old sandalwood plants that were growing in a greenhouse were selected and their roots, stems, leaves, and haustoria were collected. Fifteen plants that had the same growth were subjected to ABA treatment for 3 d and drought stress for 0 d, 3 d, and 9 d. After nine days of drought stress, in which the RWC of the soil was reduced from 70% to 32%, and after ABA treatment, each plant sprayed 200 µM of ABA [36]. Five sandalwood plants were selected after drought treatment and ABA treatment and five leaves were taken from each plant. Three biological replicates were performed. Finally, they were stored at −80 °C until RNA extraction.

### 2.2. Analysis of Cis-Reactive Elements in the Promoter

TBtools were used to extract the 2000 bp of *SaCYP736A167* gene promoter and submit the promoter sequence to PlantCARE for cis-reaction element analysis [37]. Finally, the number of each element was counted for quantity visualization.

### 2.3. Synthesis of cDNA and qRT-PCR

Plant RNA Kit (Omega Biotek, Norcross, GA, USA) was used to obtain the RNA of sandalwood, which was extracted from the plants under abiotic stress and the roots, stems, leaves, and haustoria of untreated plants. According to the instructions of the HiScript II 1st Strand cDNA Synthesis Kit (Vazyme, Nanjing, China), RNA was reverse-transcribed into cDNA.

Additionally, qRT-PCR was performed on a LightCycler 480 II Real-Time PCR System (Roche, Indianapolis, IN, USA), which is based on the SYBR Green (Sangon, Shanghai, China) method. The Primer 3 online tool was used to design specific primers of *SaAREB*s [38]. All reactions were performed for three biological replicates, as well as four technical repeats.

### 2.4. Statistical Analysis

We used the $2^{-\Delta\Delta Ct}$ method to acquire the gene expression in Microsoft Excel 2020 and analyzed whether there were significant differences in gene expression via SPSS Statistics 27 [39]. The significance of *SaAREB* gene expressions were analyzed by one-way ANOVA.

### 2.5. Genome Identification and Sequence Analysis of the SaAREB Gene Family

The sandalwood *AREB* gene family was identified by using the CDSs of nine *A. thaliana AREB/ABF* members as queries in the sandalwood genome database using the Blastp program. The e-value was set to $1 \times 10^{-20}$ for screening to reduce errors. The information on physiological and biochemical indicators was obtained by Protparam (https://web.expasy.org/protparam/, accessed on 15 April 2023) and Wolf PSORT organelles (https://www.genscript.com/wolf-psort.html, accessed on 20 April 2023) were used to predict the subcellular localization of the *SaAREB* gene [40].

### 2.6. Establishment of a Phylogenetic Tree, Analysis of the Conserved Motif, and Gene Structure Analysis

The AREB/ABF protein sequences of three plant species were used to establish a neighbor-joining tree (N-J tree) using MEGA X 10.0.5. Then, we imported the evolutionary tree into iTOL (https://itol.embl.de/itol.cgi, accessed 6 May 2023) for beautification [41]. MEME (https://meme-suite.org/meme/tools/meme, accessed 18 April 2023), as a website, can predict protein-conserved motifs; we used it to visualize the location and information of conserved motifs [42]. The gene structure map was painted by GSDS2.0 (http://gsds.gao-lab.org/index.php, accessed on 8 March 2023).

### 2.7. Chromosome Mapping and Collinearity Analysis within and between Species

To better understand the evolution of *AREB* genes across species, this study not only analyzed the tandem repetition and fragment repetition events of the *AREB* gene in *S. album* but also compared the collinearity analyses of *AREB* genes in *S. album*, *P. trichocarpa* (*Populus trichocarpa*), and *A. thaliana*. TBtools software (http://github.com/CJ-Chen/TBtools, accessed on 23 March 2023) was used to obtain and display tandem repeats and chromosome-segment repeats [43].

### 2.8. Gene Expression Analysis of SaAREBs in Different Tissues

The expression of *SaAREB*s in different tissues was acquired from RNA-seq data (Hangzhou Lianchuan Biological Technology, Hangzhou, China). The expression data of *AREB* gene family members were screened and the gene expression heatmap was drawn by the pheatmap program in R 4.2.2 software.

## 3. Results

### 3.1. The Number of Cis-Reactive Elements in the Promoter of the SaCYP736A167 Gene

*SaCYP736A167* is an important gene affecting the synthesis of sandalwood alcohol. These studies showed that α-santalene, β-santalene, α-exo-bergamotene, epi-β-santalene, and others are further catalyzed by cytochrome monooxygenase *SaCYP736A167* to produce sandalwood alcohol [10]. We found a large number of ABREs in the *SaCYP736A167* promoter using statistical analysis (Figure 1). *AREB/ABF/ABI5*, as a transcription factor, can specifically bind to the ABREs in the promoter. Therefore, to explore the effect of ABRE binding factors on sandalwood synthesis, we characterized the *AREB/ABF/ABI5* gene family of sandalwood.

### 3.2. Characteristics of SaAREB/ABF/ABI5 Gene Family Members

The CDSs of nine *AREB/ABF/ABI5* members in *A. thaliana* were used as queries in the sandalwood genome database using the BLASTP program; the e-value was set to $1 \times 10^{-20}$ for screening to reduce errors. Based on the result of the conserved domains, we identified ten *SaAREB* genes, which we named *SaAREB*1-10. In this study, their amino acid length, isoelectric point (pI), and molecular weight were determined (Table 1). The CDS length of *SaAREB*s is usually 792 bp–1365 bp, the encoded protein generally contains 263–454 amino acids, and the molecular weight ranges from 28.98 kDa to 48.28 kDa. And, according to the aliphatic index and the instability index of SaAREBs, we found that all *SaAREB* genes encoded unstable hydrophilic proteins.

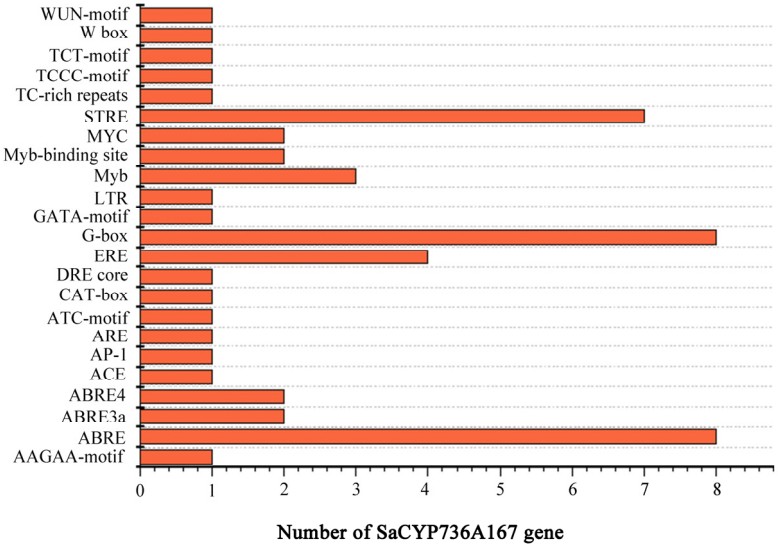

**Figure 1.** The number of cis-reactive elements in promoters of *SaCYP736A167*.

### 3.3. Evolutionary Tree, Conserved Domains, and Gene Structure of the SaAREB Gene Family

We used MEME to analyze the SaAREBs and the number of motifs was determined to be five. Based on the results, we found that only SaAREB1 and SaAREB9 were missing Motif 1 and the remaining AREBs all contained five conserved motifs (Figure 2a); among them, Motifs 1, 2, 3, and 5 contained conserved phosphorylation motifs (RXXS/T), which has been confirmed to activate AREB1 in *Arabidopsis* [44]. According to the gene structure annotation, we found that the number of introns in the *SaAREB* gene family ranged from three to five and most of them had four introns (Figure 2b).

CD Search was used to predict the conserved domains and the 10 SaAREBs we selected all contained a highly conserved bZIP (basic leucine zipper) structure. In addition, there were N-terminal C1, C2, and C3 domains and C-terminal C4 domains (Figure 3).

### 3.4. Phylogenetic Analysis of AREB/ABF/ABI5 Genes in S. album, A. thaliana, and P. trichocarpa

The *AREB/ABF/ABI5* gene family belongs to subfamily A of the *bZIP* gene family, whose genes are mainly responsible for the plant hormone response and the ability to influence stress. To better understand the evolutionary relationship between the *AREB/ABF/ABI5* in *S. album*, *A. thaliana*, and *P. trichocarpa* (*Populus trichocarpa*), we used three different methods of building trees; respectively, they are the neighbor-joining (Figure 4), maximum-likelihood, and maximum-parsimony methods (Figures S1 and S2). Through comparison, the three tree-building methods can show consistent evolutionary relationships. Based on sequence similarity and tree topology, thirty-three AREB proteins were divided into five subgroups, namely, I, II, III, IV, and V. Group I had three AREB proteins and accounted for the largest number of proteins. Group II contained only one SaAREB.

**Table 1.** Characteristics of putative genes encoding *AREB/ABF/ABI5* in *S. album*.

| Gene Name | Gene Symbol | CDS Length (bp) | Amino Acids (aa) | Molecular Weight (kDa) | PI | Conserved Domains | In Silico Prediction Wolf PSORT | Instability Index | Aliphatic Index | Grand Average of Hydropathicity |
|---|---|---|---|---|---|---|---|---|---|---|
| *Sal7G10590.1* | SaAREB1 | 1032 | 343 | 38.07 | 6.05 | C1, C2, C3, C4, bZIP | nucl | 48.27 | 65.45 | −0.711 |
| *Sal6G19740.1* | SaAREB2 | 969 | 322 | 36.18 | 6.99 | C1, C2, C3, C4, bZIP | nucl | 68.73 | 66.89 | −0.8 |
| *Sal7G09490.1* | SaAREB3 | 801 | 266 | 29.81 | 9.33 | C1, C2, C3, C4, bZIP | nucl | 45.72 | 76.69 | −0.788 |
| *Sal8G17300.1* | SaAREB4 | 831 | 276 | 31.01 | 9.24 | C1, C2, C3, C4, bZIP | nucl | 50.19 | 72.14 | −0.589 |
| *Sal7G08230.1* | SaAREB5 | 792 | 263 | 28.98 | 5.52 | C1, C2, C3, C4, bZIP | nucl | 68.88 | 73.42 | −0.653 |
| *Sal3G15050.1* | SaAREB6 | 1290 | 429 | 45.86 | 9.87 | C1, C2, C3, C4, bZIP | nucl | 56.2 | 64.76 | −0.69 |
| *Sal9G05580.1* | SaAREB7 | 1326 | 441 | 48.28 | 9.59 | C1, C2, C3, C4, bZIP | nucl | 58.95 | 62.4 | −0.875 |
| *Sal8G03390.1* | SaAREB8 | 1245 | 414 | 45.22 | 9.64 | C1, C2, C3, C4, bZIP | nucl | 60.21 | 62.87 | −0.829 |
| *Sal3G19580.1* | SaAREB9 | 906 | 301 | 33.37 | 6.88 | C1, C2, C3, C4, bZIP | nucl | 59.83 | 64.52 | −0.779 |
| *Sal5G17700.1* | SaAREB10 | 1365 | 454 | 47.84 | 8.53 | C1, C2, C3, C4, bZIP | nucl | 54.52 | 59.14 | −0.665 |

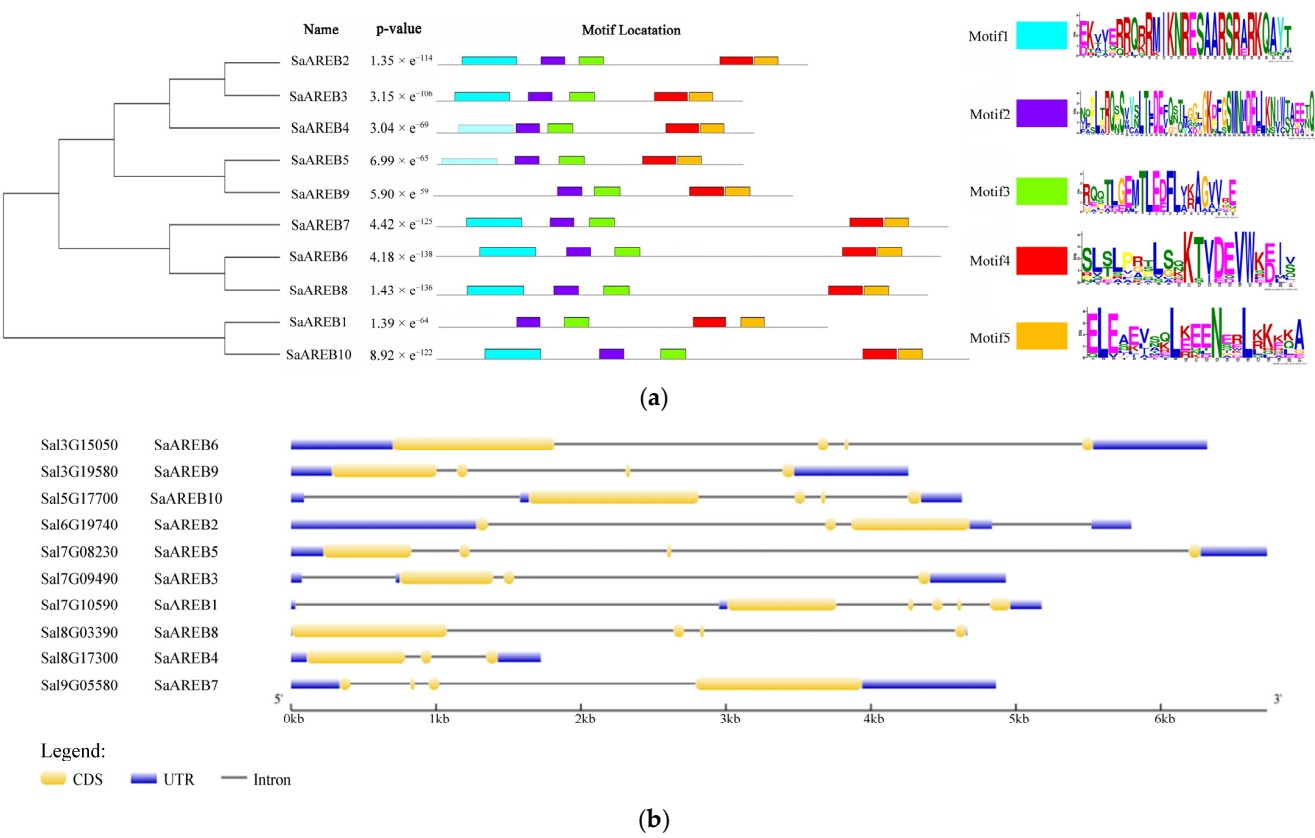

(**a**)

(**b**)

**Figure 2.** Evolutionary tree and gene structure of the *SaAREB* gene family. (**a**) Phylogenetic tree and conserved motifs based on the protein sequences of SaAREBs. Five motifs were distinguished by different colors: blue, purple, green, red, and yellow. (**b**) Genetic structure of the *AREB* gene in *S. album*. These include CDS, UTRs, and introns.

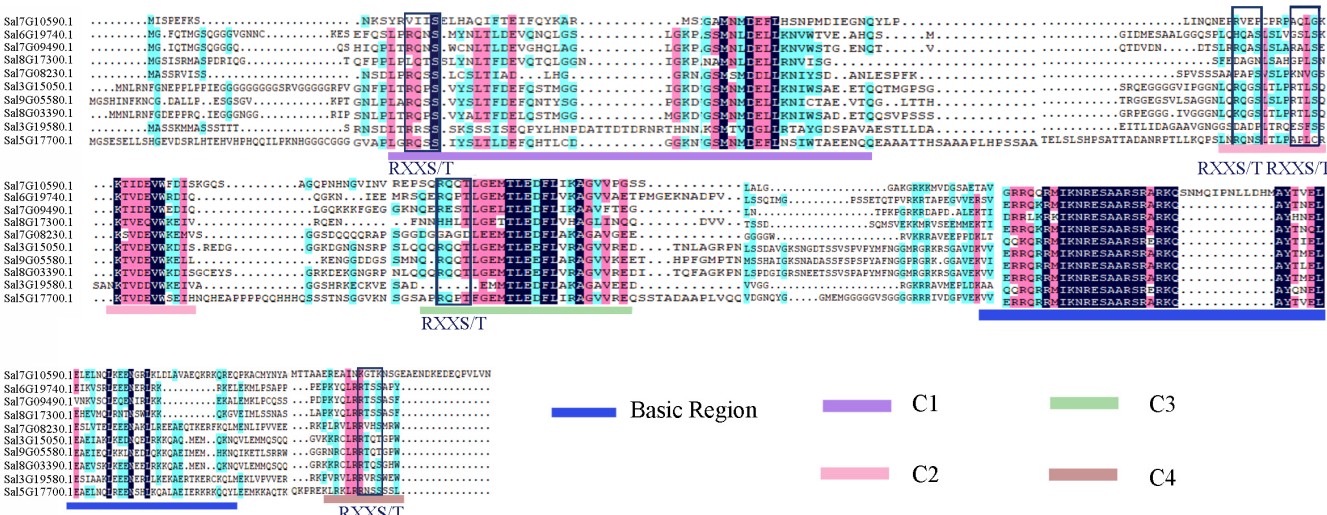

**Figure 3.** Conserved domains and phosphorylation sites of the SaAREB protein. Dark blue frames indicated potential phosphoresidues (R-X-X-S/T).

### 3.5. Chromosomal Distribution, Synteny Relationship, and Evolution of SaAREB Genes

The results of chromosome mapping showed that these genes were distributed on six chromosomes of the *S. album*. Three *SaAREBs* were located on chr7; chr3 and chr8 each contained two *SaAREB* genes and chr5, chr6, and chr9 each contained an *AREB* gene (Figure 5a). Gene duplication is a driving force for multigene family evolution; it

usually starts with whole-genome duplication, followed by fragment duplication, tandem duplication, and gene conversion. Two fragment duplication events containing four *AREB/ABF* genes were detected in the sandalwood genome (Figure 5b). However, no tandem duplication events were found in the sandalwood genome. These results suggest that fragment repetition played an important role in the amplification of the *AREB* gene family of sandalwood. To further explore the evolutionary mechanisms of the *SaAREB* family, we performed a collinearity analysis involving *S. album*, *P. trichocarpa*, and *A. thaliana*. The results showed that five and eight *AREB* genes of sandalwood had collinearity with *A. thaliana* and *P. trichocarpa*, respectively (Figure 6). Remarkably, some *SaAREB* genes were associated with at least three syntenic gene pairs, implying that these genes may play an important role in the *AREB/ABF* family during the evolutionary process.

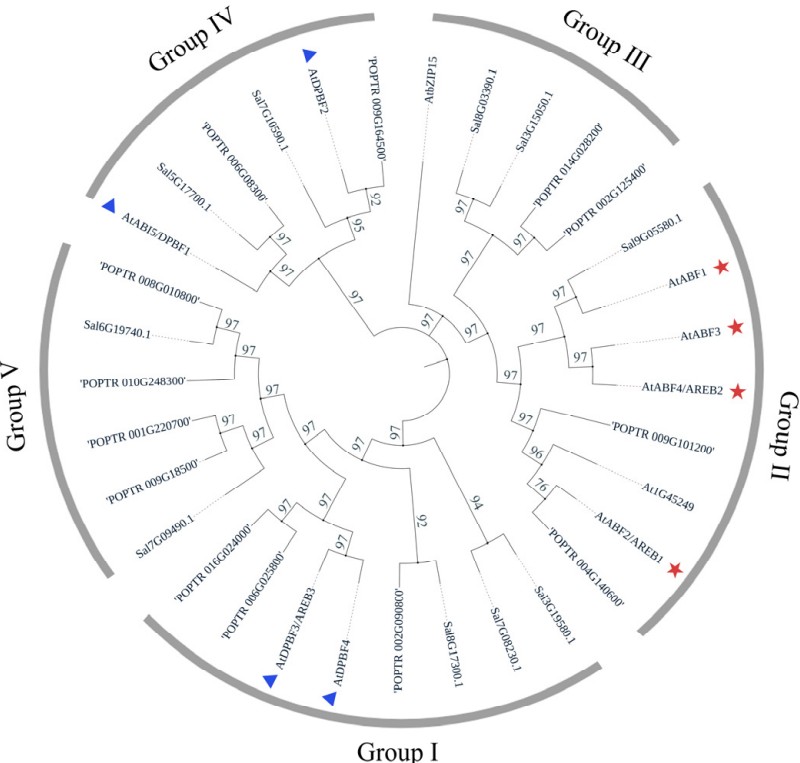

**Figure 4.** Phylogenetic relationships of AREB/ABF/ABI5 proteins in *A. thaliana*, *P. trichocarpa*, and *S. album* that were established by the neighbor-joining (NJ) connection method in MEGA X. The red asterisk indicates AtAREB/ABFs and the blue triangle indicates AtABIs.

### 3.6. Tissue-Specific Expression of the AREB Gene in S. album

To study the functions of the *SaAREB* genes, tissue-specific expression analysis was performed on 10 *SaAREB* genes and we selected four tissues: the root, stem, leaf, and haustorium. As we know, ROS signaling was necessary for the growth of sandalwood haustorium and, through the analysis of the results, we found that most of the *AREB* genes were highly expressed in the haustorium; among them, the gene that had the highest expression was *SaAREB7*, which was 22.6 times that of the root, indicating that the *AREB* gene family may be related to influence over the formation of haustorium or ROS. Moreover, *SaAREB1*, *SaAREB4*, and *SaAREB5* were also highly expressed in the stems of *S. album*; they were 2.44 times, 1.35 times, and 1.25 times the expression in the root, respectively (Figure 7), suggesting that they may be closely related to stem elongation, xylem formation or heartwood formation, and other functions of sandalwood. At the same time, we found that the expression of all *SaAREB* genes was not high in the leaves and *SaAREB6* had the lowest expression in leaves, only 0.018 times that in roots. The expressions of *SaAREB2* and *SaAREB9* were relatively high in the root.

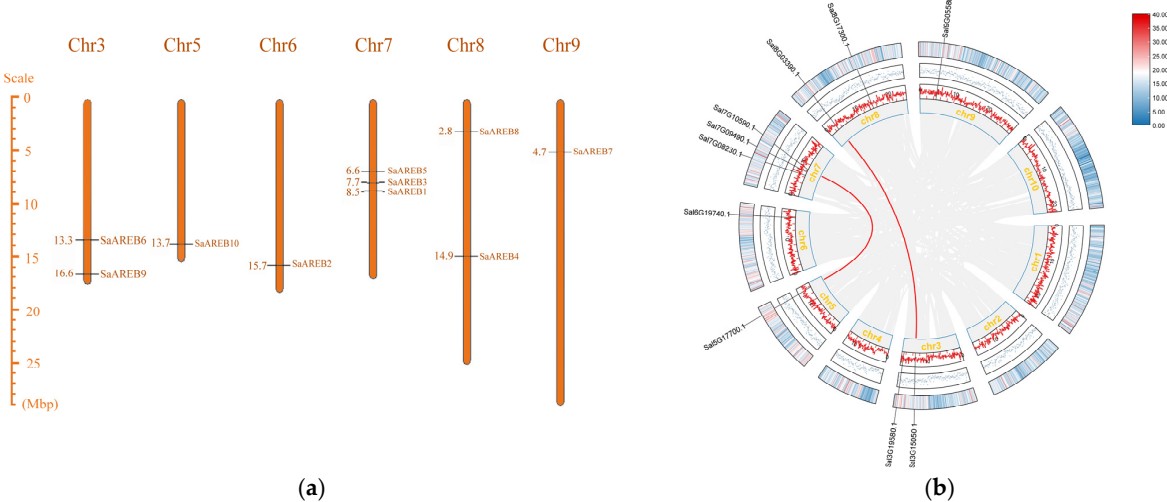

(a)                                                            (b)

**Figure 5.** The interchromosomal relationships and chromosome location map of the *AREB* genes in *S. album*. (**a**) The chromosome location map shows the location of 10 *AREB* genes and the chromosome length unit is Mbp. (**b**) The gray and red lines represent the synteny blocks in the sandalwood genome and the *AREB* repeat gene pairs, respectively. The gene density heat map is in the outermost circle, shown in blue and pink.

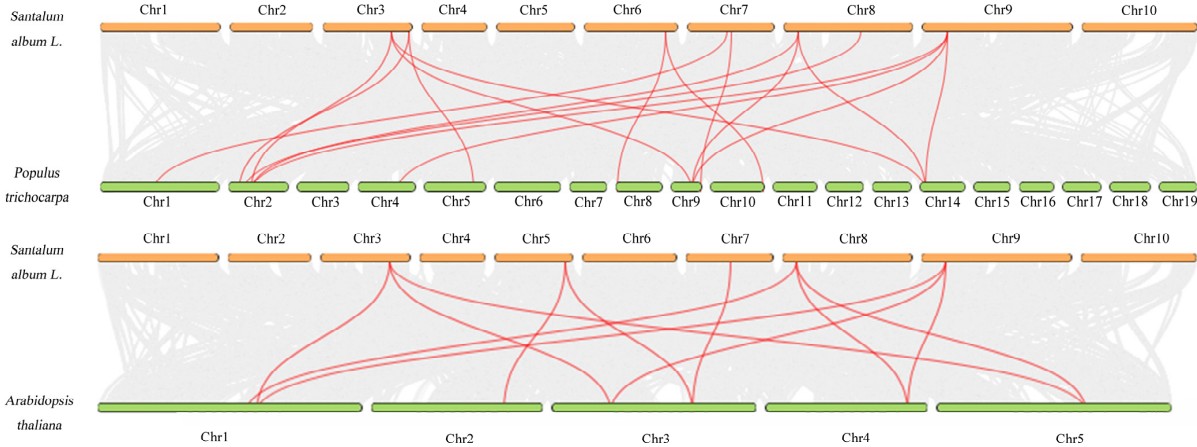

**Figure 6.** Synteny analysis of *AREB* genes between *S. album*, *P. trichocarpa*, and *A. thaliana*. Gray lines in the background indicate the collinear blocks within the *S. album*, *A. thaliana*, and *P. trichocarpa* genomes while the red lines highlight the syntenic *AREB* gene pairs.

### 3.7. Expression of SaAREB Genes under Drought Stress and ABA Treatment

To determine the expression patterns of these genes under abiotic stress, we studied the expression levels of these 10 *SaAREB* genes under drought stress and ABA treatment. According to the results, the expression levels of *SaAREB1* increased significantly after 9 days of drought stress while the expression levels of *SaAREB2*, *SaAREB4*, *SaAREB5*, *SaAREB8,* and *SaAREB10* continued to decrease after 9 days of drought stress. The expression of *SaAREB3* first decreased and then increased after drought treatment while the expression patterns of *SaAREB6*, *SaAREB7,* and *SaAREB9* were reversed (Figure 8a). This suggested that *SaAREBs* may have functions related to drought or hormone response, especially *SaAREB1*, *SaAREB3*, *SaAREB6*, *SaAREB7,* and *SaAREB9*, which positively responded to drought stress. We also found that *SaAREB3*, *SaAREB7*, and *SaAREB8* expressed significantly in response to ABA treatment (Figure 8b). In previous studies, we found that *SlAREB1*, which was homologous to the gene *AREB1* in tomatoes, was related to drought and saline—alkali stress; through the research results, we found that its homologous gene, *SaAREB7* in sandalwood, also responded to ABA treatment and drought treatment.

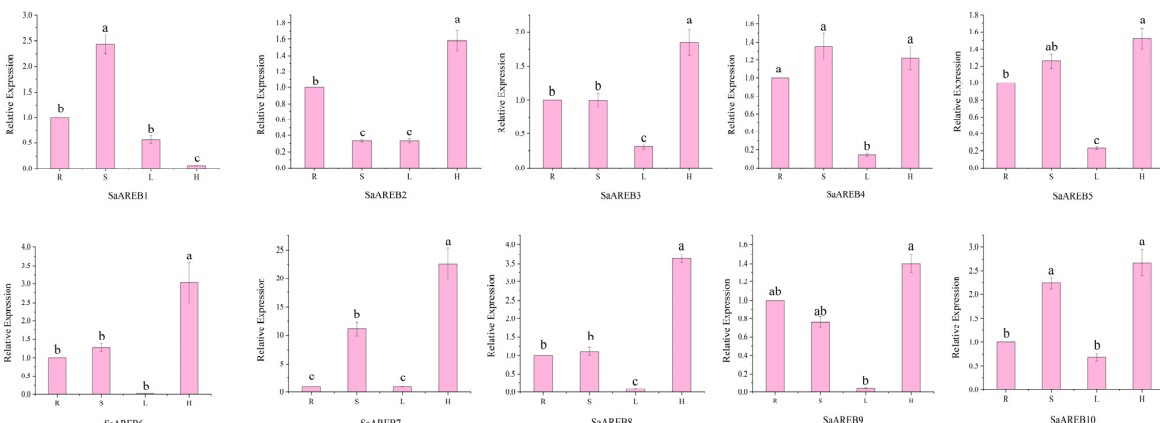

**Figure 7.** Expression analysis of ten *SaAREB* genes in four representative tissues (root, stem, leaf, haustoria) by qRT-PCR. The expression of the root was used as a control and the $2^{-\Delta\Delta Ct}$ method was used to calculate the relative expression. Error bars indicate the $\pm$ standard error (SE) of three biological replicates and a, b, c indicate the significant difference, marked with the same letter means $p > 0.05$, marked with different letters means $p < 0.05$.

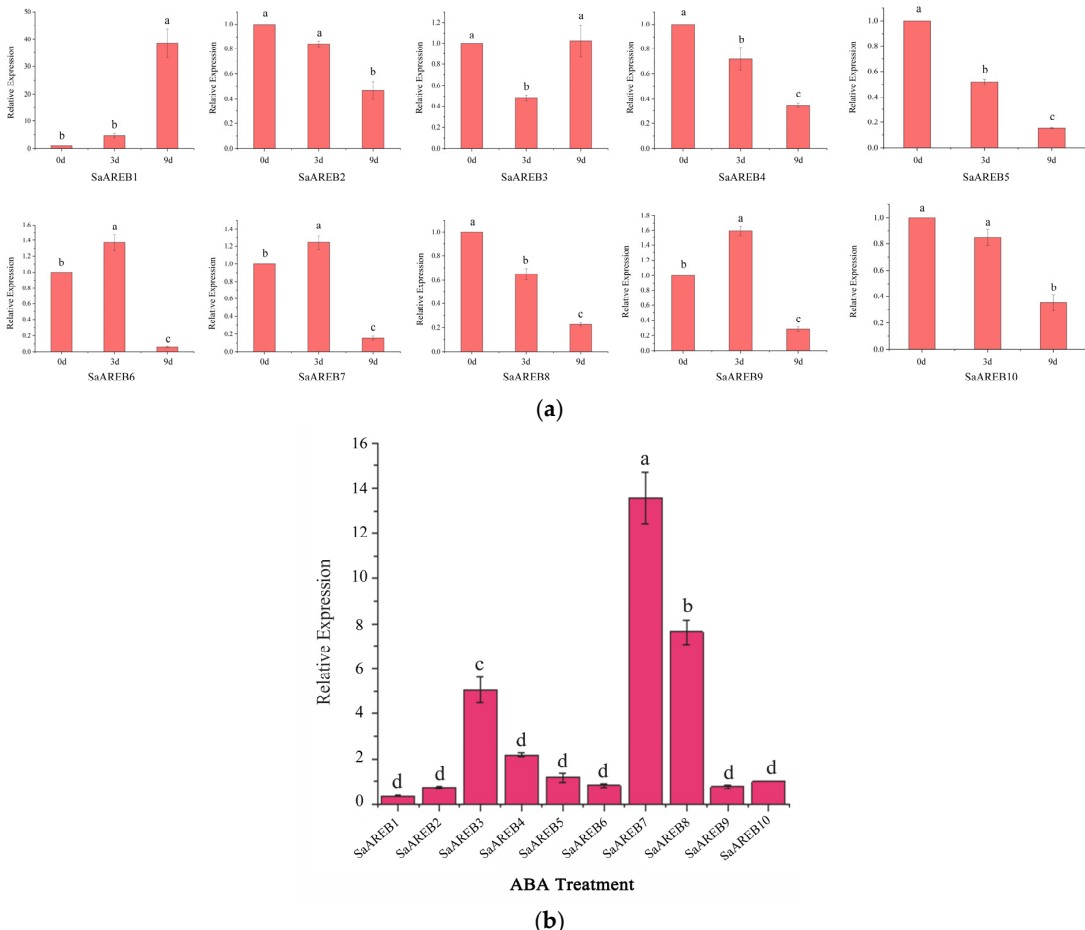

**Figure 8.** Expression profiles of *SaAREB* genes in response to abiotic stress. (**a**) Expression of sandalwood leaves after drought for 0 d, 3 d, and 9 d. (**b**) Expression profiles of the 10 *SaAREB* genes under ABA treatments. The data are presented as the mean $\pm$ standard error (SE) of three separate measurements and a–d indicate the significant difference, marked with the same letter means $p > 0.05$, different letters mean $p < 0.05$.

*3.8. Expression Profile Analysis of S. album AREB/ABF/ABI5 Genes in Various Tissues*

To understand the expression patterns of the *AREB* gene in different sandalwood tissues, RNA-seq data were used to compare the transcript abundance among different sandalwood tissues. The results showed that the expression of *SaAREB1*, *SaAREB4*, and *SaAREB10* was low or undetectable in the roots, haustoria, and leaves (FPKM < 1); whereas, the *SaAREB5*, *SaAREB6*, *SaAREB8*, and *SaAREB10* genes had much higher expression in the transition region (FPKM > 4), which suggested that they may be involved in terpenoid synthesis. Moreover, we found that the quantitative PCR results of *SaAREB5* and *SaAREB10* in the stems were consistent with the above results (Figure 9). Notably, *SaAREB6* and *SaAREB8* are homologs of *AtAREB1/ABF2* in *A. thaliana* while *SaAREB10* is a homolog of *AtABI5*. According to related studies, compared with the wild type, the *abf2* mutant of rice was more sensitive to high salt, drought, and oxidation stress [45].

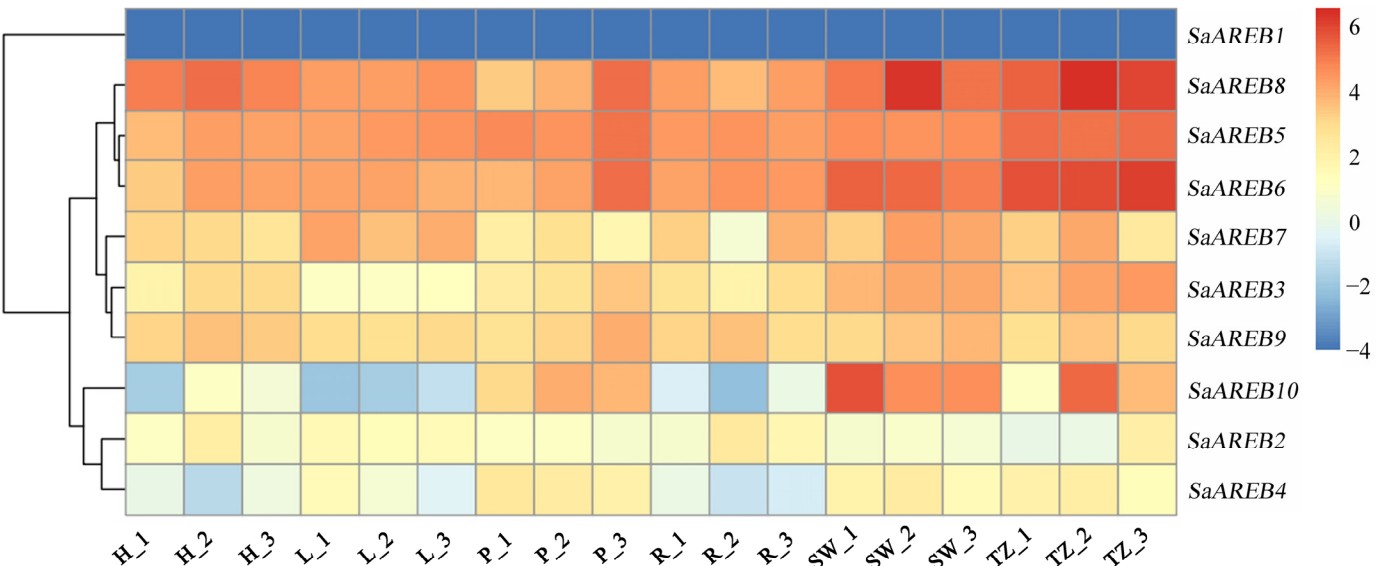

**Figure 9.** Hierarchical clustering of the expression profiles of *S. album AREB* genes in different tissues. P: phloem, H: haustorium, L: leaf, R: root, SW: sapwood, TZ: transition region.

## 4. Discussion

Sandalwood essential oil is a valuable spice with high medicinal and practical value. As the main components of essential oils, sandalene and santalol are synthesized through the MVA pathway. *SaCYP736A167,* which belongs to cytochrome p450, is a key gene in the MVA pathway for santalol synthesis. By analyzing the cis-reaction elements in the *SaCYP736A167* promoter, we found that the promoter also included a large number of ABREs.

Drought is one of the most widespread environmental factors, mainly affecting plant-water potential and turgor to cause a negative effect on agronomical, physiological, and metabolic performance [13,46]. ABA is the regulatory hormone and it plays an important role in plant growth, development, stress responses, and the synthesis of terpenoids. When plants were stressed, they affected stomatal opening and closing or the expression of defense-related genes by increasing ABA content, thus enhancing the resistance to external stress [14]. *AREB/ABF/ABI5*, as a transcription factor that can specifically bind ABREs, plays key roles in developmental processes and in the adaptation of plants to adverse environmental conditions [47,48]. However, the *AREB* gene family in sandalwood has not been reported and studied. Therefore, we characterized the *AREB/ABF/ABI5* gene family of *S. album* to explore its potential roles in sandalwood.

According to previous studies, four [19], fourteen [25], eight [49], and ten [50] *AREB/ABF* genes have been found in *A. thaliana*, *P. trichocarpa*, *C. olitorius*, and *S. lycopersicum* and, in this study, a total of ten *SaAREB* genes were identified by a local BLASTP search and HMM

model. Moreover, we found that all *SaAREB* genes encoded unstable hydrophilic proteins and that the SaAREB protein was predicted to be located in the nucleus.

The members of the *AREB/ABF* gene family are evenly distributed in subfamily A of *bZIP*, which is mainly related to hormone responses and stress responses. To explore the relevant function of the *AREB* gene, we conducted a correlation analysis of its conserved domain and found that all *AREB* genes contain four conserved domains and a terminal bZIP structure, indicating that they have functions related to those of the *bZIP* gene. Among the conserved domains of AREB/ABFs, in addition to the highly conserved basic region leucine zipper, are the C1, C2, C3, and C4 conserved domains, which include four regions containing potential phosphorylation sites [43]. These corresponding phosphorylation sites in *Arabidopsis AREB1* have been shown to be critical in regulating activation [44]. To understand the relationship between the *AREB* genes in *P. trichocarpa* and *A. thaliana*, we adopted three tree-establishment methods: maximum likelihood, minimum evolution, and neighbor joining. According to the final results, we divided the genes into five subgroups.

Gene duplication usually starts with whole-genome duplication, followed by fragment duplication, tandem duplication, and gene conversion. Understanding the evolutionary relationship of the *AREB* gene family is one of the most important steps to studying its function; so, we conducted chromosome localization and a collinearity analysis of this gene family. By chromosome mapping, we found that ten *SaAREB*s were unevenly distributed on six chromosomes, which may be related to the uneven repetition events of sandalwood chromosome fragments. Through intraspecific collinearity analysis, we detected two fragment duplication events containing four *AREB* genes in the sandalwood genome. However, no tandem duplication events were found in the sandalwood genome. These results suggest that fragment duplication played an important role in the amplification of the *AREB* gene family of sandalwood. Moreover, we found that five and eight *AREB* genes of sandalwood had collinearity with *A. thaliana* and *P. trichocarpa*, respectively, indicating that the evolutionary relationship between *S. album* and *P. trichocarpa* was closer than those with *A. thaliana*.

To preliminarily investigate the function of *AREB/ABF* and whether it responds to ABA and drought stress, we analyzed the expression levels of the roots, stems, leaves, and haustoria of sandalwood by qRT-PCR, as well as the leaves of *S. album* treated with ABA and subjected to drought stress for 0, 3, and 9 days. We found that *SaAREB1* is specifically highly expressed in stems in response to drought stress while *SaAREB7*, a homologous gene of *ABF1*, is specifically highly expressed under ABA treatment. Additionally, according to a related study, *SlAREB1* can strongly respond to drought stress and salt stress and ABA induces two identified *SlAREB* transcription factors in tomatoes [28]. In addition, we found that most of the *SaAREB* genes responded to drought stress, indicating that the involvement of the *SaAREB* gene family in the response to drought stress is universal. In *A. thaliana*, *AtABF1* is mainly involved in the response to low temperature and ABA stress; in kiwifruit, *AchnABF1* also has the same function [33,51]. It has been reported that *AREB/ABF* is widely used as a marker gene in ABA signal transduction. According to this result, *SaAREB1* and *SaAREB7* can also be used as marker genes for the ABA response in sandalwood. Moreover, sandalwood essential oil is the most valuable substance in the heartwood. To study whether the *AREB* gene is related to santalol synthesis, we processed the transcriptome data of sandalwood. According to the expression heatmap of the *AREB* gene family, *SaAREB6* and *SaAREB8*, homologous genes of *AREB1/ABF2*, are highly expressed in the phloem, transition zone, and sapwood, which indicates that they may be related to santalol synthesis to some extent. *AREB1/ABF2* in *A. thaliana* was mainly involved in ABA, drought, high salinity, heat, and oxidative stress responses and was mainly expressed in plant tissues other than seeds. In addition to *SaAREB6* and *SaAREB8*, *SaAREB5* and *SaAREB10* were also highly expressed in the sapwood and the transition zone, which was also consistent with our tissue-specific expression results showing high expression in the stem.

## 5. Conclusions

In this study, we identified 10 *SaAREB*s in sandalwood, which include the bZIP region. Through the way of multi-sequence comparison, collinearity analysis, a phylogenetic tree, and qRT-PCR, we identified and analyzed the *AREB* gene family of sandalwood and the expression pattern of the *AREB* gene between different sandalwood tissues, and under ABA and drought treatment, was investigated; we provided a way for people to understand the *AREB* gene family of sandalwood. In addition, the analysis of cis-reactive elements in the *SaCYP736A167* promoter and RNA-seq data in different tissues suggested that *AREB* might be related to the synthesis of santalol in sandalwood. We speculate that *AREB* can affect the production of santalol in sandalwood by influencing the expression of *SaCYP736A167* and that this pathway needs to be further explored in the future.

**Supplementary Materials:** The following supporting information can be downloaded at: https://www.mdpi.com/article/10.3390/f14081691/s1, Figure S1: Phylogenetic relationships of AREB/ABF/ABI5 proteins in *A. thaliana*, *P. trichocarpa*, and *S. album*, which were established by the maximum-likelihood method; Figure S2: Phylogenetic relationships of AREB/ABF/ABI5 proteins in *A. thaliana*, *P. trichocarpa*, and *S. album*, which were established by the maximum-parsimony method; Table S1: qRT-PCR primer of 10 *SaAREB* genes; Table S2: The information of the cis-reactive elements in *SaCYP736A167*; Table S3: Collinearity analysis among sandalwood; Table S4: Collinearity analysis between *S. album*, *A. thaliana*, and *P. trichocarpa.*

**Author Contributions:** X.L. and R.C. designed all of the experiments and performed the experiments. Y.C., F.Q. and S.W. conceived the project and took a sample. Y.L. and D.W. analyzed the experimental results. X.L., L.H. and S.M. wrote the paper. All authors have read and agreed to the published version of the manuscript.

**Funding:** This work was supported by grants from the National Key R&D Program of China (2022YFD2202000), the Science and Technology Research Program of Chongqing Municipal Education Commission (KJQN202101245), and the Natural Science Foundation of Chongqing, China (CSTB2023NSCQMSX0021).

**Institutional Review Board Statement:** Not applicable.

**Informed Consent Statement:** Not applicable.

**Data Availability Statement:** Not applicable.

**Acknowledgments:** We acknowledge everyone who contributed to this article.

**Conflicts of Interest:** The authors declare no conflict of interest.

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
