# Peer review of "Identification and Characterization of the AREB/ABF/ABI5 Gene Family in Sandalwood (Santalum album L.) and Its Potential Role in Drought Stress and ABA Treatment"

_forests, doi:10.3390/f14081691_

Round 1

Reviewer 1 Report

General comments

I have read the manuscript (forests -2530639). Entitle: Identification and Characterization of the AREB/ABF/ABI5 Gene Family in Sandalwood (Santalum album L.) and Their Potential Role in Drought Stress and ABA Treatment written by Xiaojing Liu et. al., for publication of forests MDPI. In this study, the author investigated the influence of AREB/ABF transcription factors on santalol synthesis by conducting the a genome-wide analysis of the AREB gene family in sandalwood. Author mainly found found all SaAREB genes encoded unstable hydrophilic proteins, and the subcellular localization prediction of SaAREBs were in the nucleus.

The overall research is well conducted, and research is obvious application potential for the readers because this research SaAREB6 and SaAREB8 were highly expressed in the sapwood and transition regions, which provide a basis for further analysis of Santalum album; ABRE/ABF/ABI5 genes in the formation of santalols. In this sense this manuscript is much more valuable. However, I found a lack of story connection and some lack of potential references (some I suggested below). Overall after I evaluate and request the author for this manuscript as a “MAJOR REVISION”.

Major Comments

1).Introduction: The introduction is well starting with Sandalwood plant importance and it high commercial and medicinal value. The drought stress is one of the challenging for the crop production and drought-caused osmotic stress significantly influences the synthesis of antioxidant, flavonoids, and secondary metabolites for the plants. 

2) Hypothesis of the study: The author presented the main aim of this study in the last section of the introduction “sandalwood AREB transcription factors which is candidate genes for the formation of sandalwood heartwood”. However, the research hypothesis is not much clear. Please mention the research hypothesis and well-connect these two parts. The hypothesis should be very clear because, without appropriate literature, questions, or hypotheses in the introduction section the entire text will be unclear.

Other comments and suggestions

3) Materials and Methods: Pease check and fix all the figures, figures are not so clear, please present pics clearly with increase the DPI and font size.

4) Line 104 (MM section): Author did not mentioned the statistical analysis section in the MM sections. Please mention it separate sub-title.

5) Line 446 (Discussion): Discussion is comparatively poorly presented. Author should be discuss the ABA section more detail and clearly. Read and refer this article as a reference https://doi.org/10.1016/j.scienta.2023.112276 . “Drought reduced hydraulic failure of the plant under the drought stress condition and reduction the plant water status by reducing the leaf water potential and sap movement that causes the negative effect in agronomical, physiological, metabolic performance, therefore ABA is the regulatory hormone, ABA plays an important role in plant growth, development and stress response.”

6) Line 532 (conclusion): The outlook and conclusion should not be repetitive in the abstract or a summary of the results section. I would love to read striking points and take home messages that will linger in the readers’ minds. What is the novelty, how does the study elucidate some questions in this field, and the contributions the paper may offer to the scientific community?

7) Line 558 (References): please double-check the citations, their style, spell check, and other grammatical errors. moreover, the author should cut the old and less matching literature and include the latest literature some of them are above.

Good Luck!

Reviewer 2 Report

1.      Modify the sentence “we explored the expression profile of …AREB transcription factors were predicted.” line no 23-24 as “we explored the expression profile of SaREB in different tissues and the effects of ABA treatment and drought treatment on AREB transcription factors were predicted.

2.      Modify abstract for more result centric, it is very introductory

3.      It is better to avoid the keywords which is already in title

4.      Briefly mention the economic and medicinal importance of Sandalwood in the introduction section

5.      Rewrite the sentence “The ABA-dependent pathway … drought stress” for better clarity line no. 57

6.      Modify the sentence “Meanwhile, drought-caused … metabolites in plants” line no 58-59 as “Meanwhile, drought-induced osmotic stress significantly influences the synthesis of secondary metabolites in plants”

7.      Briefly describe the impact of drought on Sandalwood

8.      Replace “drought treatment” with “drought stress” line. No. 108

9.      What was the concentration of ABA treatment and how it was decided?

10.  What were the moisture content of the field capacity in drought stress conditions (3 and 9 days) mention it

11.  Elaborate the section, 2.1 Plant Material and Treatment for better clarity

12.  It is better to shift Table S1 in the manuscript and put “Gene symbol” in place of “Name” in the second column

13.  Re-prepare figures 1-8 for better clarity though, they are very clear in the supplementary file. Therefore I would like to suggest preparing larger figures to avoid blurriness

14.  Table 1 is ok

15.  Write the results of gene expression analysis in terms of fold change in different tissue in a comparative manner

16.  The discussion section is written nicely and the interpretation of the results in the discussion is ok

17.  Modify the conclusion section for better soundness

Minor English corrections required

Round 2

Reviewer 1 Report

Dear Author

I have read the revised manuscript forests-2530639. Entitled: Identification and Characterization of the AREB/ABF/ABI5 Gene Family in Sandalwood (Santalum album L.) and Their Potential Role in Drought Stress and ABA Treatment for publication in forests. This is the second submission made by the author. The author addressed all the questions and suggestions that I raised the issue in the review of the original manuscript. I satisfy the author’s revisions. Author improves their hypothesis and well connected with the research objectives in this time. This manuscript improved the flow of writing, which was comparatively shallow in the original version but in this revised copy author very well addressed all the quarries and suggestions. Before accepting this manuscript if there is anything needed to be revised by the author, especially English grammar, or spell check, I request this manuscript is currently in “Minor Revision” and the author may correct any further grammatical errors (if any) the author may improve in this stage.

Thank you.
